# Population structuring of multi-copy, antigen-encoding genes in *Plasmodium falciparum*

Yael Artzy-Randrup[1]*, Mary M Rorick[1], Karen Day[2], Donald Chen[3],
Andrew P Dobson[4,5], Mercedes Pascual[1,5]

[1]Department of Ecology and Evolutionary Biology, Howard Hughes Medical Institute and the University of Michigan, Ann Arbor, United States; [2]Department of Microbiology, Division of Medical Parasitology, New York University School of Medicine, New York, United States; [3]Department of Microbiology, Division of Medical Parasitology and Department of Medicine, Division of Infectious Disease, New York University School of Medicine, New York, United States; [4]Department of Ecology and Evolutionary Biology, Princeton University, Princeton, United States; [5]Santa Fe Institute, Santa Fe, United States

**Abstract** The coexistence of multiple independently circulating strains in pathogen populations that undergo sexual recombination is a central question of epidemiology with profound implications for control. An agent-based model is developed that extends earlier 'strain theory' by addressing the *var* gene family of *Plasmodium falciparum*. The model explicitly considers the extensive diversity of multi-copy genes that undergo antigenic variation via sequential, mutually exclusive expression. It tracks the dynamics of all unique *var* repertoires in a population of hosts, and shows that even under high levels of sexual recombination, strain competition mediated through cross-immunity structures the parasite population into a subset of coexisting dominant repertoires of *var* genes whose degree of antigenic overlap depends on transmission intensity. Empirical comparison of patterns of genetic variation at antigenic and neutral sites supports this role for immune selection in structuring parasite diversity.

**\*For correspondence:**
yartzy@umich.edu

## Introduction

Many pathogen populations seem to be composed of several independent strains; this has important implications for disease surveillance and control, as well as for the identification of vaccine targets. In particular, the population genetic interactions between pathogen strains can have significant epidemiological consequences, influencing transmissibility, clinical immunity, and the duration and severity of infection. An extensive variety of microparasites have been studied in this context: protozoa, such as *Plasmodium falciparum*, *Leishmania*, and *Trypanosoma brucei*; viruses, such as *Influenza*, *HIV* and *Dengue*; and bacteria, such as *E. coli*, *Bordetella spp.* and *Neisseria meningitides* (**Tibayrenc et al., 1990**; **Gupta et al., 1994**; **Tibayrenc and Ayala, 2002**).

The conceptual challenges and significant human health implications associated with the pathogen population structure have led to the establishment of several theories for why we observe this structure in nature (**Selander et al., 1987**; **Tibayrenc et al., 1990**; **Smith et al., 1993**; **Gupta and Day, 1994a**, **1994b**; **Gupta et al., 1996**; **Tibayrenc, 1994**). In particular, while it would seem that sufficient out-crossing should always lead to random associations between alleles (**Tibayrenc et al., 1990**; **Tibayrenc and Ayala, 2002**) and preclude the persistence of 'independently transmitted strains' (**Tibayrenc, 1994**), mathematical models of competition between pathogen strains have demonstrated that immune selection can lead to the maintenance of discrete, non-overlapping,

**eLife digest** Malaria is an infectious disease that is estimated to kill more than half a million people every year, mostly young children in Africa. It is spread by mosquitos that are infected with *Plasmodium* parasites that attack red blood cells in the human body. *Plasmodium falciparum*, the species that is responsible for most of these deaths, causes malaria by entering red blood cells and releasing antigens that travel to the surface of the cells, where they change the adhesion properties. This causes the infected red blood cells to accumulate in small blood vessels, surface capillaries or the brain, which can have severe consequences for the person infected.

*P. falciparum* is particularly dangerous because of its ability to vary the antigens displayed on the cell surface: this process, known as antigenic variation, helps to maintain infections for extended periods of time by allowing the antigens to stay one step ahead of the immune system (a process known as immune escape). The origins of antigenic variation lie in the fact that each *P. falciparum* genome has a repertoire of between 50 and 60 *var* genes that code for the variability of a major antigen that is responsible for immune escape in malaria. Molecular sequencing has shown that local parasite populations contain thousands of different types of *var* genes: hence, meiotic recombination in the mosquito can create a vast number of combinations of *var* repertoires.

Artzy-Randrup et al. have developed a computational model of this highly diverse and complex system to address the following question: is a local pathogen population composed of largely random and ephemeral repertoires of these genes, or is it structured into independently circulating strains? Their model goes beyond previous models by including interactions within the local host population that arise as a result of indirect competition between different strains of the pathogen for available hosts: this competition is influenced by the history of infection and, therefore, by the patterns of immunity within the host population. Previous models included within-host processes but not these higher, local population-level interactions.

The model simulates the dynamics of all the unique combinations of *var* genes in a population of hosts, and shows that even with high rates of reproduction, the parasite population self-organizes into a limited number of coexisting strains: the distinct *var* repertoires of these strains only weakly overlap, suggesting that the immune response of the host population has been partitioned into distinct niches. By investigating genetic variation at both antigenic sites and regions of the genome that do not code for antigens, Artzy-Randrup et al. suggest that immune selection—the selection imposed on *var* repertoires by the build up of specific immunity at the population level—plays a central role in structuring parasite diversity.

The new model should lead to a better understanding of the epidemiology of *Plasmodium* and other pathogens that work in similar ways, including *Trypanosoma brucei* (sleeping sickness), *Borellia burgdorferi* (Lyme disease) and *Giardia lamblia* (gastroenteritis), and help with global efforts to eliminate malaria and other diseases.

antigenic repertoires—even when antigenic variability is driven by several unlinked genes that undergo recombination (**Gupta et al., 1996**; **Gupta and Anderson, 1999**). An inherent challenge in testing 'strain theory' is to determine whether observed empirical patterns in pathogen population structure result from immune selection, or alternatively, from simple demographic processes (i.e., uneven relatedness among individuals in the pathogen population that can result from random genetic drift, particularly within the complex context of transmission dynamics) (**Buckee et al., 2008**, **2011**).

*P. falciparum* is responsible for the deadliest type of malaria in humans. The complexities of its population genetics, in terms of the variable and often enormous levels of sequence variation observed at antigenic sites within local populations, are intimately tied to both its pathology and its ability to escape immune detection. While this genetic complexity provides significant challenges to malaria research and control, it also provides an opportunity to extend existing strain theory in a way that can be tested empirically. Specifically, an important extension of earlier strain theory is to consider the effects of immune selection in populations of pathogens whose antigen-encoding loci are: (a) multi-copy and recombining (i.e., located on different chromosomes of a haploid genome), and (b) under-going antigenic variation via sequential, mutually exclusive expression within an infected host. It is not clear whether a stable strain structure would emerge under the sequential expression of multi-copy

genes. Consideration of this type of system provides a more complete framework for understanding antigenic structure and its epidemiological consequences in a broader variety of pathogens; it also offers the opportunity to distinguish the effects of immune selection from those of neutral demographic forces, providing testable predictions for these two alternative mechanisms.

We describe an agent-based model that extends strain theory to address the effect of immune selection acting on multi-copy antigen genes; we use the *var* gene family of *P. falciparum* as the basis for this model. The *var* genes encode *P. falciparum* erythrocyte membrane protein 1 (PfEMP1)—the major surface antigen of the parasite's blood stages that binds to host endothelial cell receptors to allow the parasite to sequester within various host tissues (*Miller et al., 2002*; *Scherf et al., 2008*). *Var* genes are composed of multiple Duffy-binding-like (DBL) and cysteine-interdomain-rich (CIDR) domains that encode binding functions for specific endothelial receptors. DBLα, is the only domain found in nearly all *var* genes (*Miller et al., 2002*), and hence is the most informative molecular marker of *var* gene diversity. Each parasite genome has up to 60 *var* genes, and differential sequential expression of these genes allows the parasite to evade the host immune response and establish a chronic infection. This clonal antigenic variation sensu stricto (*Miller et al., 2002*; *Scherf et al., 2008*) extends the duration of infection to facilitate transmission to the mosquito. Not only do *var* genes influence transmission, but specific *var* genes are associated with severe malarial disease and specific PfEMP1 binding phenotypes (*Deitsch et al., 2009*; *Chan et al., 2012*; *Claessens et al., 2012*; *Lavstsen et al., 2012*; *Warimwe et al., 2012*).

Population studies have shown that individual *P. falciparum* genomes have distinct repertoires of *var* genes (*Barry et al., 2007*; *Scherf et al., 2008*; *Chen et al., 2011*) that extensively recombine (*Freitas-Junior et al., 2000*; *Taylor et al., 2000*) during sexual recombination in the mosquito (*Paul et al., 1995*). The immense sequence diversity observed among *var* genes and *var* repertoires has been attributed to high rates of meiotic (and possibly also mitotic) recombination. These high rates of recombination presumably allow for very low linkage disequilibrium among *var* loci. Our modeling framework considers the dynamics shaping the enormous potential diversity of genomic repertoires of *var* genes by tracking all possible *var* repertoires, given a diverse pool of genetic variants. We explore whether structured parasite populations resulting from the selection pressures exerted by competition among distinct *var* repertoires for hosts, mediated by cross-immunity, can emerge.

## Results

We model the dynamics of multiple parasite genomes that simultaneously circulate in a large host population, continuously tracking the formation of new genomes through recombination, mutation and immigration events (*Paul et al., 1995*; *Bruce et al., 2000a*, *2000b*). Reliable numerical results require the simulation of explicit transmission events, as well as complete knowledge of the exposure history of each individual host with respect to past and present infections by different parasite genomes. Thus, an agent-based model is formulated that follows each host from birth to death, while recording their exposure history and the specific immunity each gradually gains as a result of infections (for a detailed description of the model see the 'Materials and methods' section).

The simulated host population is continuously exposed to an immense diversity of distinct *var* repertoires, since new combinations of *var* genes are continuously generated through meiotic recombination events. As a conservative example, consider the number of potential configurations arising from the combinatorial possibilities of 6 loci with 50 *var* genes in the pool: more than $3 \times 10^7$ repertoires! Despite this large potential diversity, simulation results show that only a few dominant *var* repertoires circulate and persist at any given time, effectively keeping *var* repertoire diversity orders of magnitude lower than is potentially possible given the number of *var* genes present in the host population (*Figure 1*). We find that epidemiological dynamics significantly limit the diversity of coexisting dominant genomes under a broad spectrum of transmission intensities. This pattern persists in the long-term, far out-living the influence of initial conditions, and persisting through occasional changes in the identity of the dominant *var* repertoires (*Figure 1*). Thus, on epidemiological (and longer) time scales, most of the parasite population is structured into a limited number of distinct *var* repertoires.

We find that the parasite population structure exhibits punctuated turnover: brief episodes of restructuring result in a new set of dominant *var* repertoires. Despite the changing identities of the dominant *var* repertoires, the overall diversity of repertoires present and the individual gene frequencies remain remarkably constant (*Figure 1*, lower panels). These transient phenomena punctuate periods of stasis with stable population structure. In essence, although there are an effectively infinite

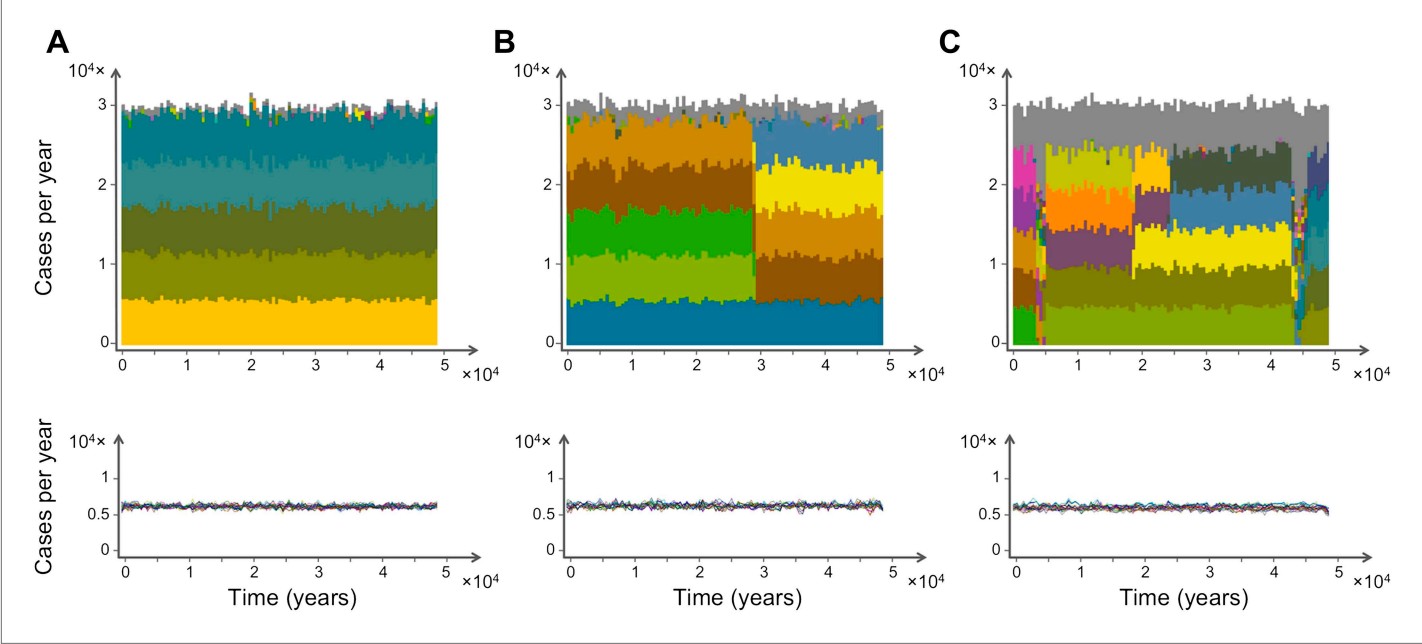

**Figure 1**. Characteristic time series demonstrating the emergence and maintenance of population structure under three levels of recombination probabilities, 0.001 (a), 0.01 (b) and 0.1 (c), while keeping all other parameters equal. Top: stacked histograms of the number of genome transmission events per year of the different circulating repertoires, each represented by a different color. All *var* repertoires sampled less than 50 times are pooled and plotted in grey. Bottom: The corresponding number of transmission events of each *var* gene per year (each gene plotted in a different color). As recombination increases, population turnover is more rapid, still showing well-defined population structure. Although dominant genome identities may change in time, the level of transmission per gene is relatively constant and hardly influenced by turnovers on the strain population level. The relative fraction of short-lived genomes increases with recombination (grey area), and although the total number of transmission events per year is not influenced by recombination rate, cases of transmission of de-novo genomes is higher for higher levels of recombination. Parameters: $b = 18$, $G = 30$, $g = 6$, $\alpha = 15$, $H = 10,000$, $\omega = 0.0005$, $\tau = 1$.

number of niches in host immunity space, *var* repertoires appear to exploit this space by partitioning it into a number of discrete and partially overlapping niches. In addition, these repertoires appear to quickly adapt and achieve some optimum after perturbations arising from the stochastic nature of the underlying demographic and transmission processes. The episodes of restructuring become more frequent as recombination rates increase, but the diversity of dominant *var* repertoires remains low throughout these semi-stable states. Indeed, the maintenance of structure is found to be robust even under very high levels of recombination (***Figure 1***). The emergence of pathogen population structure is robust to alternative formulations of our multi-locus model that do not consider sequential expression, or formulations that consider randomly ordered sequential expression. In addition, episodes of restructuring can also be observed when expression is not sequential.

The level of diversity remains orders of magnitude lower than that potentially allowed by the *var* gene pool itself. However, as transmission intensity increases, the overlap between distinct dominant *var* repertoires also increases, with completely discordant genomes only appearing at the lower end of the spectrum (see ***Figure 2***). To capture the characteristic features of different population compositions we defined four metrics: $N*$, measuring the average number of successful transmission events per year, $V*$, measuring the average number of de-novo genomes transmitted per year, $R*$, measuring the diversity of *var* repertoires, and $F*$, measuring the degree of *var* overlap between distinct repertoires (see 'Materials and methods'). We find that as transmission intensity (biting rate) increases, all four measures increase as well, and that epidemiological dynamics clearly constrain *var* repertoire diversity and structure parasite populations, with the degree of overlap determined by epidemiological parameters (***Figure 3A–D***). Our findings are robust to different host population sizes (from 1,000 to 100,000 individuals), different gene pools sizes (from 10 to 500), and different repertoire lengths (from 2 to 30) (***Figure 3***).

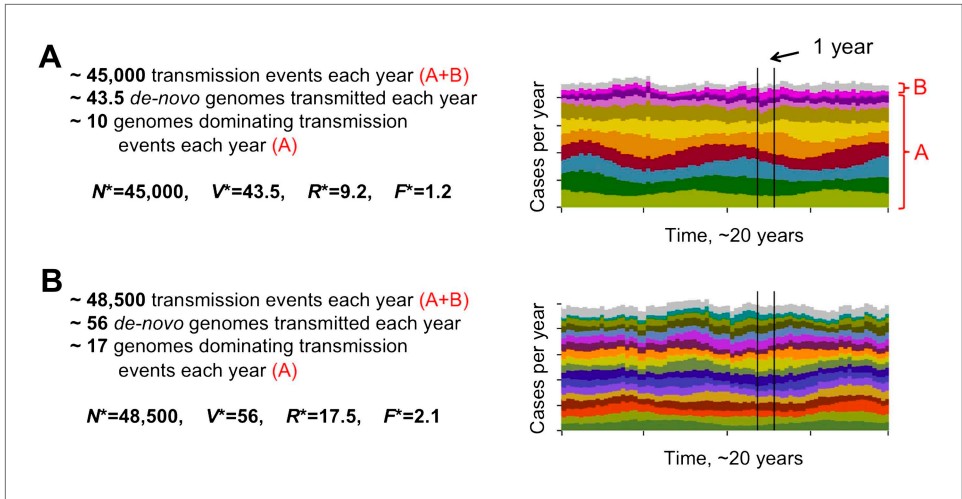

**Figure 2**. Different degrees of distinct *var* repertoire overlap with transmission intensity. The number of genome transmission events per year for the different circulating repertoires are represented by different colors as stacked histograms similar to those of *Figure 1*. Emerging populations are typically characterized for low transmission intensities (**A**), by a low number of dominating genomes with a low level of antigenic overlap ($R^* = 9.2$, $F^* = 1.2$, see 'Materials and methods'); and for high transmission intensities (**B**), by higher diversity and higher antigenic similarity ($R^* = 17.5$, $F^* = 2.1$). The fraction of long-lived genomes dominating transmission events is labeled by A in both figures, and the fraction of short-lived genomes, by B (these are grouped into one single number and plotted in gray). Parameters: $b = 8.5$ and $b = 9.5$, $G = 50$, $g = 6$, $\alpha = 15$, $H = 20,000$, $\omega = 0.0005$, $\rho = 0.01$, $\tau = 1$, $\kappa = 1$ year.

The model suggests that areas that experience different transmission intensities will exhibit 'characteristic' epidemiological signatures (*Figure 4*). Importantly, the gain of immunity is found to depend on the degree of repertoire overlap (i.e., the number of shared *var* alleles) among the different dominant genomes and not just on total prevalence (*Figure 5*). This is an important difference with compartmental transmission models that consider superinfection but have no explicit treatment of *var* genes and repertoire diversity.

Two recent large-scale studies aimed at assessing the extent of local reservoirs of *var* genes in five epidemiologically and geographically distinct *P. falciparum* populations (***Barry et al., 2007***; ***Chen et al., 2011***) provide us with new knowledge on the distribution of *var* genes in parasite genomes. These new data clearly show that the frequency distribution of distinct *var* genes, or sequence 'types' (defined by the level of sequence identity within the DBLα domain) is significantly different from what we would expect if sequence types were equal in frequency and uncorrelated (*Figure 6*). In addition, the data show significant nonrandom associations among many *var* DBLα types within the sampled isolates (*Figure 6*).

These patterns in observed *var* repertoires may alternatively result from neutral demographic processes: that is, individual mosquitoes bite discrete hosts, and thus, inevitably sample the host population unevenly. This limits 'random mating' and thus recombination between parasite genomes. Hence, even under neutral forces, it is possible for parasites to differ in their relatedness to one another. Under this alternative assumption we expect these neutral demographic processes to shape the diversity at all loci to the same extent (after controlling for differences in physical linkage and recombination rate between loci). Evidence of more extensive 'strain structure' at antigenic, immune-selected sites compared to neutral sites suffices as proof that selective forces are at work, and of these, immune selection is the most parsimonious explanation.

We compare patterns of variation at immune-selected sites to neutral sites in a variety of ways (see 'Materials and methods'), and we propose this as an effective empirical test of strain theory. Our test is conceptually similar to empirical analyses of strain theory undertaken in the past (***Buckee et al., 2010***), but in this system, there is the opportunity for a more powerful test since the multi-copy nature of *var* genes allows us to examine patterns of variation at many independent (i.e., physically unlinked)

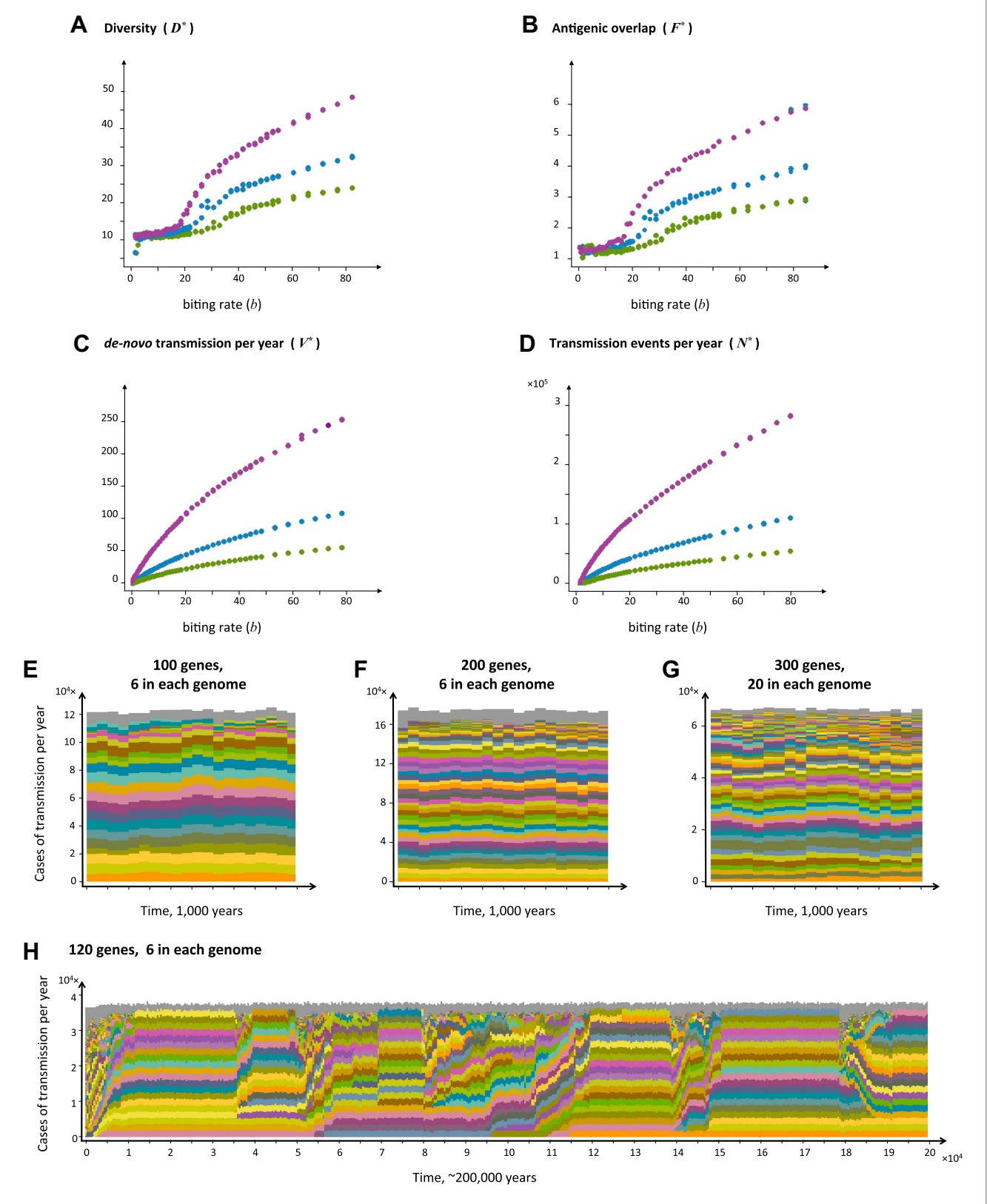

**Figure 3**. Top four panels: the metrics characterizing population structure for three host population sizes as a function of transmission intensity (see 'Materials and methods'). Green, blue and purple represent $H$ = 5,000, $H$ = 10,000, and $H$ = 25,000, respectively. (**A**) A measure of diversity capturing the number of dominating genomes ($R^*$, in the y-axis) as a function of $b$, (**B**), a measure of average antigenic similarity between genomes
*Figure 3. Continued on next page*

*Figure 3. Continued*

($F^*$, in the y-axis) as a function of $b$, (**C**) the average number of novel repertoires generated and successfully transmitted each year, and (**D**) the total number of successful transmission events each year. As transmission intensity increases, the number of dominating genomes increases, and in accordance, the degree of antigenic similarity increases as well—however the number of distinct repertoires that are maintained in the system at epidemiologically relevant frequencies remains extremely low relative to the total number of distinct repertoires that can be crated through recombination. Note the differences in magnitude; while the number of de novo repertoires entering the system is 10-fold higher then the observed diversity, the number of successful transmission events that these dominating repertoires are involved in is on the order of $10^6$. For extremely low biting rates, in a narrow window of parameter values, diversity decreases abruptly (**A**). This reflects a different dynamic regime in which dominant strains replace each other in a fluctuating manner and population structure does not apply. Parameters: $G = 50$, $g = 6$, $\alpha = 15$, $\omega = 0.0005$, $\rho=0.05$, $\tau = 1$, $\kappa = 1$ year. Bottom four panels: characteristic time series demonstrating structured parasite populations for larger gene pools ($G$) and longer sequence lengths ($g$): (**A**) $b = 18$, $G = 100$, $g = 6$, $\alpha = 15$, $N = 20{,}000$, $\omega = 0.0005$, $\rho=0.001$, $\tau = 1$. (**B**) $b = 18$, $G = 200$, $g = 6$, $\alpha = 15$, $H = 20{,}000$, $\omega = 0.0005$, $\rho=0.001$, $\tau = 1$. (**C**) $b = 18$, $G = 300$, $g = 20$, $\alpha = 15$, $H = 20{,}000$, $\omega = 0.0005$, $\rho=0.001$, $\tau = 1$. (**D**) $b = 12$, $G = 120$, $g = 6$, $\alpha = 15$, $H = 10{,}000$, $\omega = 0.0005$, $\rho=0.005$, $\tau = 1$, $\kappa = 1$ year.

genetic sites. Currently, the dataset with the most extensive sampling of diversity at both *var* and microsatellite loci within a local population consists of a set of 30 single infection isolates from a local population in Papua New Guinea for which we have sequences for a portion of the *var* DBLα domain and the alleles for 12 microsatellite loci (*Barry et al., 2007*; *Chen et al., 2011*).

Because we expect alleles at the same locus to anti-correlate, and because it is unknown which DBLα alleles share a common locus, we only analyze positive correlations. We measure the positive linkage disequilibrium coefficients (*D*) for the presence/absence states for all unique pairs of microsatellite or *var* DBLα alleles. High *D* values indicate a correlation between alleles within isolates, and they are maximized at intermediate allele frequency (see and 'Materials and methods'). For the *var* DBLα dataset there are 1288 distinct pairs of alleles with positive *D* values that exceed a two-tailed significance threshold of 0.05 (*Hedrick et al., 1978*). Given the number of tests performed, under the null hypothesis of no significant correlations between alleles, we would expect only 310 pairs to exceed this threshold. For the microsatellite dataset there are 146 distinct pairs of alleles with positive *D* values that exceed the significance threshold and under the null hypothesis we would expect only 49. Therefore, both datasets show an enrichment of positive correlations among allele pairs, and thus, significant linkage disequilibrium. The *D* values of the *var* DBLα dataset are higher than those of the microsatellite dataset ($p_v \ll 0.0001$ by Mann–Whitney test), and we argue that this cannot easily be explained by allele frequency differences or differences in the extent of physical linkage among sites (see 'Materials and methods').

Another important observation is that, at least at this level of resolution, *var* DBLα repertoire relatedness is not a function of microsatellite repertoire relatedness. *Figure 6E* shows the relatedness of the *var* DBLα repertoires of an isolate pair as a function of the relatedness of their microsatellite repertoires. The absence of any correlation suggests the possibility that *var* DBLα variation is shaped by other forces than the neutral demographic forces shaping microsatellite variation.

We also consider only the most informative *var* types in isolation by identifying highly correlated sets of *var* types. We then tested the relatedness of microsatellite alleles among the isolates containing these networks of correlated *var* types. The networks of correlated *var* types were constructed from pairs of *var* types exhibiting both high linkage disequilibrium and correlation coefficients (i.e., *D* and *r*). *Figure 6D* represents one of the largest of such fully connected networks (with $D > 0.0748$ used as the cutoff), however this complete network is only observed in two isolates in the dataset. The microsatellite repertoires of these two isolates are less related than average for the whole dataset, so at least in this case, high microsatellite relatedness does not explain the existence of these highly correlated *var* types. Additionally, smaller subsets of correlated *var* types are often found in isolates with very dissimilar microsatellite alleles. These results suggest that the existence of highly correlated *var* types is not always explained by high levels of underlying genome relatedness. Moreover, when we use the same cutoff *D* value to look for fully connected networks of correlated microsatellite alleles, the largest of such networks comprises only four alleles as opposed to six (six pairwise connections as compared to 15).

Although the sample size of this dataset is too small to identify anything but very strong correlations between intermediate frequency *var* types (*Brown, 1975*), these analyses support a possible role

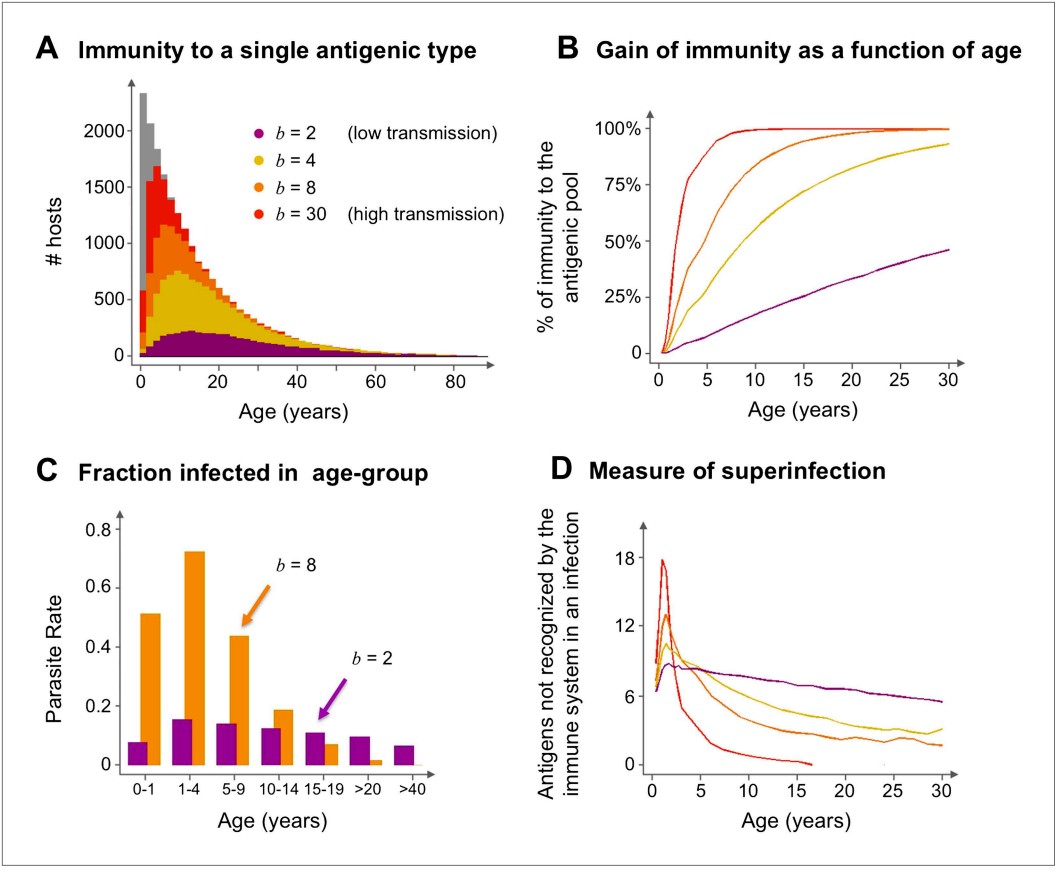

**Figure 4**. Age profiles of immunity and infection: (**A**) In color are the number of hosts immune to a single arbitrary antigen in different age classes superimposed over a histogram representing the entire age distribution of hosts (in gray). The level of immunity in each class is shown for four different transmission intensities ($b$ = 2, 4, 8 and 30, where $b$ is the average number of biting events per host per year). (**B**) The average level of immunity to the general antigenic pool as a function of age. It can be seen in (**A**) and (**B**), that at higher levels of transmission intensity, exposure to the diversity of antigenic types typically shifts towards the younger age classes, and hosts gain high levels of immunity to a majority of the variants earlier on in life. (**C**) A measure of parasite rate demonstrated by the fraction of infected hosts in different age groups (see also **Gupta and Day, 1994b**) for 2 transmission intensities ($b$ = 2 and 8). (**D**) A measure of superinfection demonstrated by the average number of antigens an infected host is naïve in each age class. Under high-transmission regimes, we characteristically observe high levels of superinfection (i.e., relatively long durations of infection) and high parasite rates in the very young age classes, after which immunity is rapidly gained to most of the antigenic variants circulating in the population. In contrast, host populations in regimes of very low transmission, characteristically exhibit similar levels of parasite rates in all age groups, and immunity is gradually gained throughout life, with infection spanning all age classes. Parameters: $G$ = 40, $g$ = 6, $\alpha$ = 15, $H$ = 20,000, $\omega$ = 0.0005, $\rho$=0.01, $\tau$ = 1, $\kappa$ = 1 year. Transmission events were sampled at a frequency of 1:100, and the host population was sampled every 10 years at a ratio of 1:100.

for immune selective forces in structuring the variation at an important class of antigen-encoding loci in *P. falciparum*. More extensive datasets will be needed for definitive conclusions, but these are now within reach as next generation sequencing methods make it feasible to consider sampling whole communities of parasite genomes to assess *var* population structure.

## Discussion

In accordance with predictions from the original 'strain theory', our model constrains total repertoire diversity to levels that are orders of magnitude lower than is potentially possible given the diversity of the gene pool. This reduction in diversity occurs because coexisting parasites are structured into a few variants with largely discordant repertoires. In a departure from previous strain theory, our model

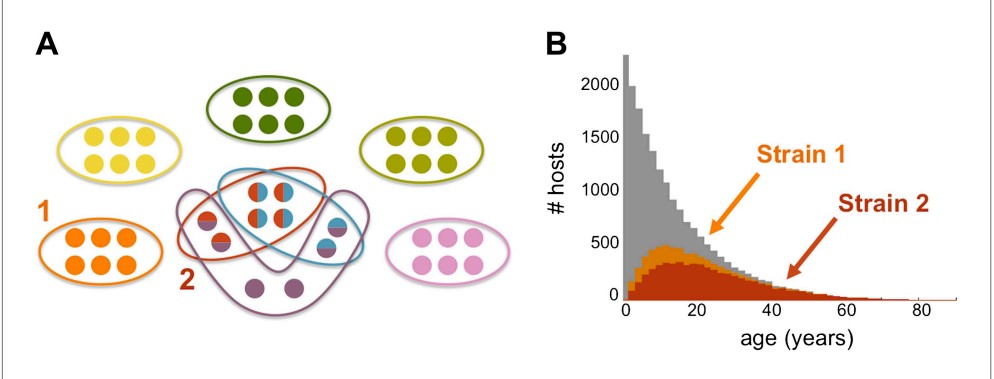

**Figure 5**. Strain specific immunity and transcending immunity. (**A**) Diagram demonstrating the partition of 40 genes from the general pool (represented by circles) between 8 dominating strains (each marked with a unique color, and composed of a subset of 6 genes). Five of these strains are antigenically unique as they have no overlap with any of the other strains, two of these strains share all of their genes with other strains (red and light blue), and one of the strains has a partially unique antigenic profile sharing 66% of its genes with other strains (purple). (**B**) The number of hosts immune to strains 1 and 2 (orange and red, respectively) for each age class superimposed over the host age distribution (grey). Immunity to strain 1 is strain specific and only gained through exposure to this particular strain due to the uniqueness of its antigenic profile. By contrast, immunity to strain 2 is strain 'transcending' and acquired also by exposure to other strains with which it shares genes. It follows that hosts are exposed to strain 1 at a higher level and at younger ages, than those for strain 2. Parameters: $b = 3$, $G = 40$, $g = 6$, $\alpha = 15$, $H = 20,000$, $\omega = 0.0005$, $\rho = 0.01$, $\tau = 1$, $\kappa = 1$ year.

predicts that the different dominant repertoires can exhibit partial overlap; the extent of the overlap increases with transmission intensity and this, in turn, influences observed epidemiological patterns (particularly age-serology profiles).

The specific immune profile of each individual host effectively structures the population into distinct sets of 'patches' or 'resources' that define an immune environment, ultimately causing immune selection to act on the parasite population. These distinct sets of patches are formed and conserved even when immunity to the *var* alleles is shorter-lived. Observations of individuals becoming susceptible to severe infection at later stages of their life (see for example (*Dhingra et al., 2010*)), may be in part a consequence of short-lived immunity. Under this assumption, hosts in our model regain protection by exposure to one of the dominant strains circulating in the host population at high frequencies, and do so at a sufficiently high rate to preserve strain population structure except at extremely low transmission intensity. In the event that a host is infected by one of the low frequency genomes that are continuously being introduced (and equally quickly vanishing), it is highly unlikely that this successful transmission event is sufficient for invasion or even replacement of the currently dominating strain community.

Discordant repertoires that exhibit partial immunological/antigenic overlap blur the distinction between the acquisition of strictly strain-specific immunity and the acquisition of strictly strain-transcending immunity (*Gupta and Day, 1994a*, *1994b*) (see caption of *Figure 5*). This underscores the role of strain diversity structure, rather than diversity per se, in determining observed epidemiological patterns and their response to intervention. By reducing antigenic overlap between strains, vector control efforts will result in infection at an older age and a more gradual acquisition of protection to infection, even in the absence of changes in *var* gene diversity. The model in its current form does not differentiate infection from clinical disease, thus we cannot explicitly address the impact of control on the severity of symptoms. However, if the number of *var* genes that a host has previously encountered relates to the degree of protection against clinical disease, we would expect more severe symptoms to occur at younger ages under high transmission settings, and in contrast, more homogeneous malaria incidents are likely to occur over a broader range of age-classes when transmission is lowered (*Figure 4D*). Investigation of these patterns awaits the formulation of models that include functional, non-antigenic, differences between strains, as well as more empirical understanding of the role played by *var* genes associated with more severe disease. Future work should thus extend our

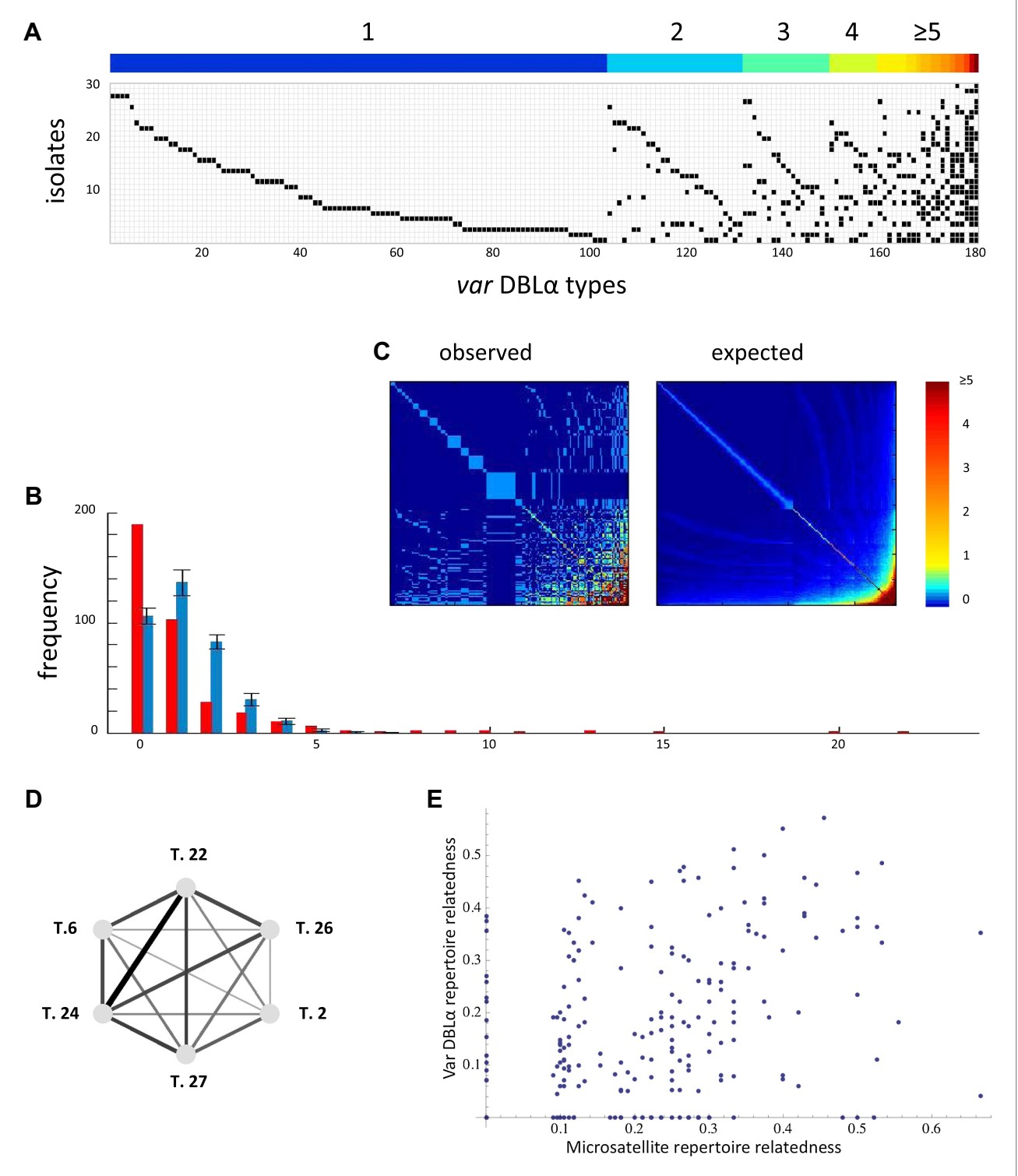

**Figure 6**. Nonrandom organization of *var* DBLα types between 30 sampled parasite isolates from Amele, Papua New Guinea as published in (***Chen et al., 2011***). (**A**) Presence–absence matrix showing the distribution of *var* types in isolates: Each row corresponds to an individual parasite isolate and each column to a distinct *var* type. The color bar indicates the number of distinct isolates each *var* type was identified in. (**B**) The frequency distribution of the number of isolates in which each *var* type was identified, obtained by assuming a total *var* richness of ~369, as estimated in (***Chen et al., 2011***). The observed distribution for the data (red) and the expected distribution when assuming all *var* types circulate at equal frequencies (blue) were found to be significantly different ($p_v<0.0001$ using a K–S two sample test). Note the enrichment for both very rare and very common

*Figure 6. Continued on next page*

*Figure 6. Continued*

*var* types in the data. (**C**) Observed and expected co-occurrence of *var* DBLα types. In each matrix cell (*i,j*) is the number of times *var* types *i* and *j* were found in the same isolate. The co-occurrence of *var* types found in multiple isolates is significantly higher than for the null expectation under the assumption of linkage equilibrium among all *var* types ($p_v < 0.01$) (***Artzy-Randrup and Stone, 2005***) (the expected co-occurrence matrix conserves both the observed number of *var* types in each isolate and the observed frequency distribution of different *var* types (***Chen et al., 2011***)). (**D**) A network of allele pairs with both high *D* and *r*. The *D* values are among the highest pair-wise comparisons in the dataset (using a cutoff of $D > 0.0748$), and the *r* values range from 1 to 0.58, and are reflected by the weight of the edges. Based on sequence features (***Bull et al., 2007***), and the fact that genomic organization of *var* sequence groups appear to be conserved across globally distributed isolates (***Kyes et al., 2007***), it is very unlikely that *var* type 22 (being UpsA-like) is physically linked to the other alleles in the network (since they are all UpsB/UpsC-like). (**E**) The relatedness of the *var* DBLα repertoires of an isolate pair as a function of the relatedness of their microsatellite repertoires. Relatedness is defined as the number of *var* types shared between the two isolates, divided by the average number of *var* types identified per isolate for that pair of isolates.

model to consider functional, non-antigen-encoding differences between *var* genes (***Bull et al., 1999***; ***Miller et al., 2002***; ***Scherf et al., 2008***; ***Buckee et al., 2009***; ***Claessens et al., 2012***; ***Severins et al., 2012*** for an example of a recent model applied to questions on the evolution of virulence). Ultimately, ways to represent the effects of pathogen diversity implicitly within a standard epidemiological framework, that might also be more easily parameterized, should also be investigated.

The goal of malaria eradication is to reduce the reservoir of malaria infection in humans to zero. ***Chen et al. (2011)*** have pointed to the importance of *var* genes in malaria surveillance, due to their influence on both the duration of infection and the degree of superinfection, and they demonstrated the existence of an immense diversity of *var* genes and limited *var* repertoire overlap in Africa, even in sites with low malaria prevalence. Consequently, they have proposed that *var* typing be used to complement prevalence measures for molecular surveillance of malaria control and elimination efforts. Our models support this proposal as they show that decreasing transmission provokes nonlinear changes in the dominant *var* configurations. Importantly, our results demonstrate that there is not a simple relationship between *var* diversity and repertoire diversity. Indeed, the '*var* strain' model suggests that vector control strategies that reduce transmission will not always drastically reduce *var* gene diversity, because a reservoir of parasite genomes with diverse (highly non-overlapping) *var* repertoires will remain. This insight from the model provides a plausible explanation for why it is so difficult to eliminate malaria: non-overlapping *var* diversity will create significant inherent difficulties in reducing the potential of the malaria parasite to reinfect the same hosts. It also suggests that *var* surveillance might provide a good molecular means of evaluating control. If the end game is to be achieved in malaria control, then the mechanisms creating and maintaining antigenic diversity warrant further investigation in both modeling and empirical studies.

In conclusion, the results of our model on the dynamics of population structure in *P. falciparum* as defined by the *var* multi-gene family, have broad implications for the possible emergence of similar structure in many important human and animal pathogens that have major surface antigens encoded by multigene families, including bacterial, protozoan and fungal pathogens (***Deitsch et al., 2009***).

## Materials and methods

### Epidemiological model

Each parasite is composed of a set of exactly *g var* genes, and the genomes are distinguishable from each other by their unique *var* combination. The overall diversity of *var* genes circulating in the host population is assumed to originate from a general predefined pool of *G* variants (***Barry et al., 2007***; ***Chen et al., 2011***; ***Recker et al., 2011***). Hence, the total number of distinct *var* repertoires is $G!/(G-g)!$. In the beginning of a simulation we generate five random repertoires by sampling variants from this pool, and we infect 10 random hosts with each of these repertoires. The size of the overall host population is *H*. Throughout the simulation, variants from this pool are sampled, either by the mutation of one *var* gene to another, at probability ψ, or through an immigration event, where at rate ζ a random host in the population is infected by a *var* repertoire randomly sampled from this pool. While mitotic recombination within *var* genes may play an important role in generating additional *var* diversity, we do not explicitly include this process here, because for the purposes of this model, and given the high levels of diversity already present in the pool, it would play a role essentially

equivalent to mutation or immigration. All our results are robust to variation in these two parameters.

During the course of an infection, a parasite expresses all of its *var* genes, and these are responsible for encoding PfEMP1 variant surface antigens (VSAs) (*Gupta et al., 1994*; *Miller et al., 2002*; *Scherf et al., 2008*; *Tibayrenc and Ayala, 2002*). After the host is exposed to each of these VSAs, it gains life-long immunity that is specific to each of the antigens (*Bull et al., 1998*; *Recker et al., 2011*). We assume that the total duration of infection in a susceptible host lasts a fixed time ($\kappa_{yrs.}$), and that in partially immune hosts, the actual duration decreases proportionally to the number of *var* genes that have been expressed in earlier infections (*Bruce et al., 2000a*, *2000b*; *Giha et al., 2000*). It is recognized that the gain of immunity during the course of an infection is gradual (*Ewers, 1972*; *Miller et al., 1994*), and we capture this feature by considering that the *var* genes composing an infecting parasite are mutually and exclusively expressed in a fixed sequential order. Hence, at each stage of an infection, the host has already acquired specific immunity to all the VSA's encoded by *var* genes expressed up to that point in time. As a result, immunity to each infecting parasite is gained gradually. The infection history of each host is tracked in time and determines its immune status (*Bull et al., 1999*; *Bruce et al., 2000a*, *2000b*; *Giha et al., 2000*; *Buckee et al., 2009*; *Doolan et al., 2009*). This effectively structures the population of hosts into distinct sets of individual 'patches' or 'resources' that define an immune environment, which ultimately causes immune selection to act on the parasite population.

The specific rules of infection constitute a static within-host sub-model and an implicit representation of within-host dynamics, which allowed us to implement the population transmission model without explicitly representing the mechanisms that give rise to the sequential expression of different VSAs within the host. As addressed in the main text, these mechanisms remain poorly understood, although multiple within-host models for sequential waves of parasitaemia have been developed (*Frank, 1999*; *Recker et al., 2004*; *Frank and Barbour, 2006*).

An infected host can harbor multiple simultaneously co-infecting parasite genomes, and each has the potential of being transmitted to another host through a mosquito biting event. When such an event takes place, each of the infecting parasites has a probability, $\tau$, of successfully being transmitted from the infected host to the mosquito, making it possible for several different genomes to be transmitted in a single biting event (see *Figure 7A*). However, in our model we assume that the actual transmission probability of each of the genomes, is divided by the total number of co-infecting genomes at the time of the biting event. Therefore, in effect, there is a cost to co-infection that is consistent with empirical evidence for a density-dependent regulating mechanism on parasite abundance within the host (*Bruce et al., 2000a*, *2000b*; *Bruce and Day, 2002*, *2003*) (see *Figure 7B*). To demonstrate the robustness of our results under different assumptions of the within-host representation, we also considered four alternative rules governing the dynamics of co-infection. For these different rules we find similar results to those presented here. The alternative rules for within host dynamics of multiple infections are: (1) *Co-infection, no cost*: during the course of co-infection, there is no cost on the transmission probability of each of the infecting genomes, and hence the transmissibility of each remains, $\tau$, identical to what it would be, had there been no other infecting genomes. (2) *Co-infection, full cost*: during the course of co-infection, the transmissibility of the earliest infecting genome is not compromised by the later infections and stands at $\tau$, all later infections are not transmissible. When the earliest infection is cleared it is the next one that becomes fully transmissible. (3) *Mutually exclusive infections*: here, co-infections are not permitted; while a host is infected with one infection, it is not susceptible to any other. (4) *Delayed super-infection*: a host can harbor multiple parasite genomes, but only one actively infects the host, with transmissibility $\tau$, while the rest remain latent until the active parasite genome is cleared. After that, one of the latent parasite genomes begins its active state until it is cleared, and so on.

Because the model is stochastic and formulated in continuous time, the implementation of the numerical simulations relies on the well-known Gillespie algorithm (*Gillespie, 1977*) for three of the processes; biting (responsible for transmission), host death, and immigration of new parasite genomes, whose respective rates are $b$, $\alpha$, and $\zeta$. The algorithm considers that these processes are Poisson and determines, based on these rates, the timing of each of these events. We further consider that the population size is constant in time, such that in the case of a death, a new susceptible host is born, replacing the previous host that died. We assume that this new host has no history of exposure, nor any acquired immunity. At a 'biting event', a vector bites two randomly chosen hosts sequentially, if the first host is actively infected by one or several parasite genomes at that time, there is a probability

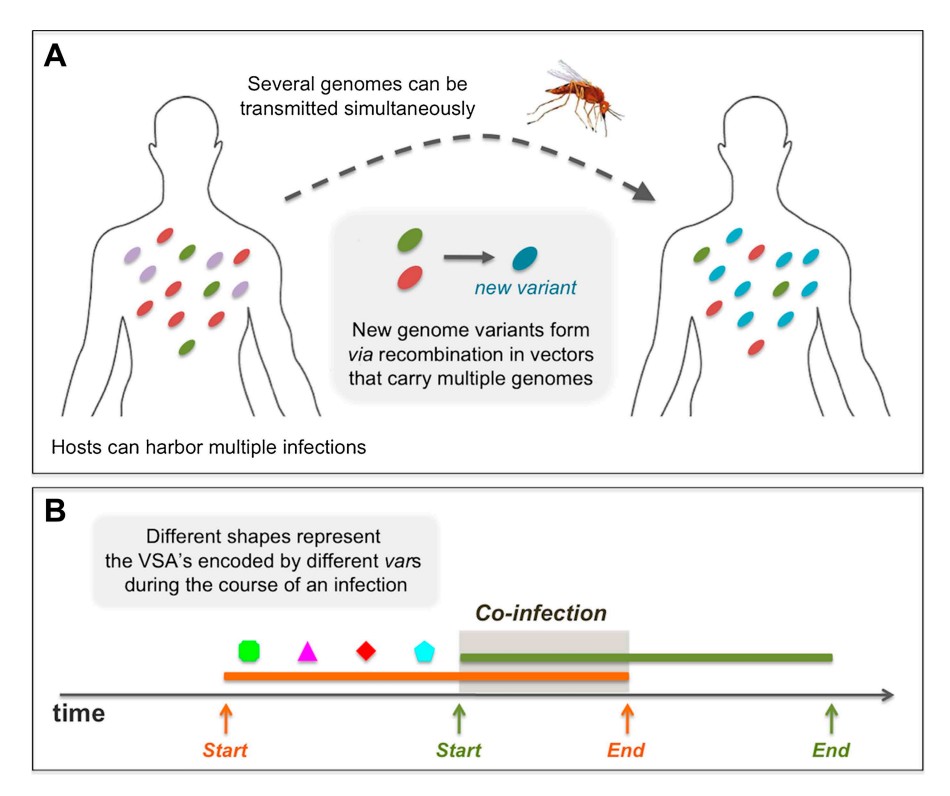

**Figure 7**. Diagrams showing processes that take place during the course of a simulation: (**A**) in a biting event, there is likeliness that multiple parasite genomes will be transmitted simultaneously. In these events, at probability ρ, the genomes recombine to create new parasite genome variants, which then infect the receiving host. (**B**) A sketch of the rules governing the dynamics of infection and co-infection. By sequentially expressing different *var* genes during the course of an infection, host immunity is gradually gained to the PfEMP1 variants (VSA's) associated with the infecting parasite genome. When the host is exposed to an additional genome, this parasite begins expressing its *var* genes sequentially, as it would, had it been infecting the host on its own. Transmissibility of each of the infecting parasites during such periods of co-infection decreases according to the multiplicity of infection at the time.

for each of these to be transmitted to the second host. As noted earlier, this probability depends on transmission probability, $\tau$, and on the number of co-infecting parasite genomes. A parasite genome that is transmitted from one host to another, has a probability $\psi$ of mutating, such that one of its randomly chosen *var* genes is replaced by a randomly chosen variant from the pool. This new genome is then the one transmitted to the second host. When two or more parasite genomes are successfully transmitted from the first host to a mosquito, there is a given probability, ρ, that recombination occurs, resulting in a new *var* repertoire that is then transmitted from the mosquito to the second host. In this case, the novel combination is formed by randomly sampling a subset of *var* genes from each of the parent repertoires. Hence, when ρ>0, this effectively leads to a continuous reshuffling of *var* repertoires circulating in the host population. Given the extensive number of possible *var* combinations, these recombination events often introduce de novo parasite genomes not yet present in the infected host population. Epidemiologically, after the introduction of a de novo genome, this parasite may successfully be transmitted multiple times, possibly leading to an epidemic, and persisting in the host population for extended periods of time. Alternatively, however, it may be unsuccessful in its transmission, persisting only for a short duration of time before stochastically going extinct.

In our numerical experiments, we consider different levels of transmission intensity by specifically varying the biting rate. Thus, throughout the paper, variation in transmission intensity refers to variation in this parameter (and therefore, in the so-called entomological inoculation rate, or EIR).

## A measure of diversity

$R*$ is calculated using Simpson's Diversity index, $R* = 1/\sum_{i=1}^{S} p_i^2$, where $S$ is the number of unique genomes responsible for transmission events in a window of time, and $p_i$ is their relative proportion. The benefit of using Simpson's index over alternative diversity indices is that it has low sensitivity to species at low abundance (in this case genomes), and gives most weight to the most abundant ones. This is appropriate in our case as we are aiming to identify the number of dominating unique genomes, and are not interested in the general richness ($S$), which in our case can be immense.

## A measure of antigenic overlap

The calculation of $F*$ is also based on Simpson's Diversity index, with a focus on the distribution of genes between transmission events, such that: $F* = \sum_{j=1}^{G} \left( 1/\sum_{i=1}^{S} p_i^2 \right) / G$. As before, $S$ is the number of unique genomes responsible for transmission events in a window of time, and for each gene $i$ from the gene pool, $p_i$ is its relative frequency in each of the transmitted genomes. Hence, a gene that is not shared between dominating genomes will be represented at high frequency at a single genome (and the diversity will be close to 1). In contrast, the frequency of a gene that is shared by two dominating genomes will be split between both those two (and the diversity will be close to 2). We then average the degree of overlap between all genes.

## Molecular marker of *var* diversity

Because DBLα is the only domain found in nearly all PfEMP1 variants (*Miller et al., 2002*), the sequence encoding the central portion of this domain provides the most informative molecular marker of *var* gene diversity (*Freitas-Junior et al., 2000*; *Taylor et al., 2000*; *Barry et al., 2007*; *Chen et al., 2011*). Two DBLα tags are considered to represent two distinct *var* 'types' if their nucleotide sequence identity is less than 96%.

## Microsatellites as neutral markers

Microsatellites offer an ideal and practical means to construct a null hypothesis reflecting the demographic forces shaping diversity because these sites are predominantly neutral, and their diversity can be explained by a mechanism unrelated to immune selection (i.e., slippage). In contrast, single nucleotide polymorphisms or single feature polymorphisms, because they are predominantly found in multi-copy antigenic genes in *P. falciparum* (*Jiang et al., 2008*), do not offer a good basis for a null hypothesis.

## Relatedness

Relatedness is defined as the number of *var* types shared between two isolates, divided by the average number of *var* types identified per isolate for that pair of isolates.

## Linkage disequilibrium coefficient *D*

$D$ is maximized when allele frequencies are 0.5, so $D$ also indicates the amount of information upon which the coefficient is based (e.g., if two alleles appear only once and in the same isolate, their $D$ will be low despite the fact that they are in perfect correlation with one another). The higher $D$ values for *var* DBLα alleles versus microsatellite alleles cannot be attributed to differences in allele frequencies since the *var* DBLα alleles have lower frequencies than the microsatellite alleles, and thus, $D$ values biased to lower magnitudes. While the higher $D$ values from the *var* DBLα dataset could be due to the influence of the close physical linkage of a small subset of pairs of *var* loci (*Deitsch and Hviid, 2004*), it seems somewhat unlikely because the great majority of *var* loci are located very far from other *var* loci in the 3D7 genome (*Deitsch and Hviid, 2004*), and *var* loci have been shown to coincide with recombination hotspots (*Mu et al., 2010*; *Jiang et al., 2011*) in a genome that is already characterized by remarkably high underlying rates of recombination (*Conway et al., 1999*; *Volkman et al., 2006*).

## Acknowledgements

We thank Sunetra Gupta and Caroline Buckee for discussions on the subject, and Daniel Zinder for implementation of the simulation code. YAR and MMR are Howard Hughes Medical Institute postdoctoral Associates and MP is an Howard Hughes Medical Institute Investigator.

# Additional information

## Competing interests

MP: Reviewing Editor, *eLife*. The remaining authors have declared that no competing interests exist.

## Funding

| Funder | Grant reference number | Author |
|---|---|---|
| The Wellcome Trust Programme | Grant 041354 | Karen Day |
| National Institutes of Health/ National Institute of General Medical Sciences | GM061351-07 | Karen Day, Donald Chen |
| National Institutes of Health Ruth L. Kirschstein National Research Service Award | F32AI071765 | Donald Chen |
| McDonnell Foundation | | Andrew P Dobson |

The funders had no role in study design, data collection and interpretation, or the decision to submit the work for publication.

## Author contributions

YAR, Conception and design, Analysis and interpretation of data, Drafting or revising the article; MMR, Conception and design, Analysis and interpretation of data, Drafting or revising the article; KD, Generated and provided the expertise on the empirical data, Drafting or revising the article; DC, Generated and provided the expertise on the empirical data; APD, Conception and design, Drafting or revising the article; MP, Conception and design, Drafting or revising the article

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
