## [Author Response]

*1) The Introduction could be expanded to provide more thorough background for non-specialist readers. Please provide a more detailed paragraph on the biology of the var genes, including their domain structure (as it will help the reader understand the data section more easily), before explaining what the model does. We would recommend expanding and clarifying the background sentences “Individual Plasmodium genomes…among var loci”, before describing the model and its findings*.

We have now expanded the background information on the *var* genes in the Introduction.

*2) The authors observe episodes of restructuring leading to punctuated turnover. It is clear from the exposition that this is a dynamically different process to the cyclical strain structure exhibited by the deterministic model under intermediate levels of immune selection. If possible without extensive further analysis, are the authors able to comment on the following? To what degree does the sequential expression itself contribute to this phenomenon? In other words, is this behavior likely to be observed in a stochastic multi-locus model without sequential expression? Or is the instability inextricably linked to sequential expression*?

In extensions of our work we have shown that episodes of restructuring with punctuated turnover do not depend on sequential expression. This phenomenon was observed in stochastic multi-locus models, both with, and without, sequential expression. However, we cannot yet say if these episodes are more likely to occur when expression is ordered.

We should also note that the likeliness of these episodes may vary when considering different within host rules, but nonetheless, in all the cases we discussed here, episodes of restructuring did take place.

We have reported the robustness of this phenomenon to relaxing the assumption of sequential expression in the Results section (see end of the third paragraph in this section).

*3) It is not obvious that “This important insight provides a plausible explanation for why it is so difficult to eliminate malaria”. However, the predicted failure of transmission control strategies to reduce var diversity may have important ramifications for the reduction of disease, given that PfEMP1 is now firmly established as the principal target of clinical immunity. Would the authors feel comfortable speculating on the differential effects of reducing transmission on disease and infection*?

We have rewritten that sentence in the Discussion to better clarify what we meant. We have also added a paragraph in the Discussion on the effect of control. We have only briefly speculated on the consequences for disease vs. infection with respect to their age distribution. To better support this paragraph, we have replaced one of the plots in Figure 4 to illustrate age distributions, and added Figure 5 to better illustrate the results that were already in the text regarding the change in the overlap of repertoires with transmission intensity.

*4) Please provide some explanation on how well the model works under the assumption that immunity is not life long given recent evidence that malaria may well kill adults as well as children (Dhingra et al, Lancet 2010). This needs to be addressed only in the Discussion*.

We have added a discussion on short-lived immunity and the reference by Dhingra et al. in the second paragraph of the Discussion to indicate that our results are robust to modifying the assumption of lifelong immunity.

*5) Parts of the Materials and methods are difficult to follow and parameters such as nh are not always clearly defined. Please make an effort to clarify. Two-letter parameter names, e.g., gp, sl, nh, can look like products. Please consider changing these. Why is the total number of distinct var repertoires given by gp!/sl! instead of gp!/((gp-sl)! sl!)*?

We have replaced these parameter names with the following: G represents the general gene pool size, g is the number of genes in each parasite genome, and H is the total population size. We think this new notation helps to make this section clearer.

The total number of distinct *var* repertoires should be G!/(G-g)! (earlier this was gp!/sl!). This is because in our model we assume the order of expression is fixed and this order matters when counting the different repertoires. Hence, for a group of g genes there are exactly g! different strain types. These different strains are composed of identical genes, but the particular order of expression of each of these can have significance, particularly in the presence of other strain types that have partially overlapping genes. More generally, we are considering the number of ordered arrangements of n objects taken from N different objects; this is given by N!/(N-n)!

Note that fixed order is not a necessary condition for our results. We simulated cases where for each infection event, the order of expression of the particular genome being transmitted would be reshuffled with a certain probability. In the extreme case when this probability is 1, for a group of g genes there would effectively be exactly 1 strain type because the particular order of expression would not matter any more. For these different cases, structuring of the parasite population was also seen to emerge.